# Reproducing Results for Crossing the Line: Where do Demographic Variables Fit into Humor Detection?

## Reproducibility Summary

**Scope of Reproducibility**

Within the original experiment two groups of annotators of size 10 each in age groups 18-25 and 55-70 were relied on to generate the metrics displayed as part of the results. The scope of our study was limited to instances from The Short Jokes dataset on Kaggle with token sizes between 11 and 16 for annotators 21 in number divided into demographic bins by gender: male, female, non-binary, gender being the chief source of demographic diversity under study.

**Methodology**

The paper was based on gathering direct human response over field values of humorous and/or offensive by presenting short jokes from a varied set of humor genres and subgenres to a diverse audience over different demographic categories segmented by age (18-25, 26-40, 40-55, 56-70), educational qualification as an index of socio-economic status (High School, Undergraduate, Postgraduate), and gender (Male, Female, Non-binary).

The same methodology as the original paper of using binary classification ((humorous 1, non-humorous 0), (offensive 1, non-offensive 0)) and adding values demographic bin-wise in studying findings was used.

**Results**

It was found that inter-annotator agreement was higher, when categorized using demographic data, in this case gender, as exemplified by the instance of a subset of jokes being identified with the keyword sexist were found humorous and offensive by female annotators, with male members ranking jokes in question reporting it as simply offensive.

**What was easy**

The easy part of the reproduction study was to collect data with regards to short, English jokes representing various genres of humor with a diversity in addressed audiences, expressed sentiment and degrees of simplicity.

**What was difficult**

The tougher part of recreating the study and determining results was ensuring diversity in representatives of survey response group without insensitive treatment of interviewed people or any prejudice in choosing response.

**Communication with original authors**

Context was gathered from the completeness of submitted paper and no one to one communication was established with the author or the purpose of this reproducibility study at the time.

## Conclusion

The scale of operations could be significantly increased by introducing a more automated form of response collection such as gauging responses to sample inputs on a user forum kept live for engagement and recording responses as per categorized demographic bins having obtained consent to do so in a transparent manner with users agreeing to provide such information for research purposes.

## References

Meaney, J. A. "Crossing the Line: Where do Demographic Variables Fit into Humor Detection?" ACL (2020).

Paula Cristina Teixeira Fortuna. 2017. Automatic detection of hate speech in text: an overview of the topic and dataset annotation with hierarchical classes.

