# OpenReview forum: "Reproducing Results for Crossing the Line: Where do Demographic Variables Fit into Humor Detection?"
_ML_Reproducibility_Challenge/2021/Fall — Reject_

### Official Review · Reviewer_4SdW · 2022-03-08
**Scarce report left a lot to be analyzed**

**Rating:** 2
**Confidence:** 4

**Review:**

The reproducibility authors (RA) produced a reproducibility report on the paper “Crossing the Line: Where do Demographic Variables Fit into Humor Detection?”.
My understanding is that the RA tried to reproduce the social experiments presented in the paper.
However, I cannot be certain of this as the report is extremely vague.

For instance, the reproducibility report does not describe the context of the study. Furthermore, the absence of a clear description of which claim was being tested hurts the readability of the report.
Because of its extreme conciseness, the report contains only very scarce information about what experiment was performed. For instance, I deduce from the methodology section that the RA performed a sociological study, but there is no information on how the participants were gathered, or details on the methodology of the study itself.
The sections on “what was easy” and “what was difficult” are very imprecise, and do not give any detail on how the difficulties were overcome. A recommendation is made regarding the collection of new data, but not on the level of reproducibility of the report.
Finally, there is no metric reported, no plot to illustrate the discussion and no table to ground the conclusions.

Overall, the report was far too imprecise for me to have gained knowledge about the level of reproducibility of the original paper.

---

### Official Review · Reviewer_ibtR · 2022-03-09
**Demographic Variable in Humor Dataset**

**Rating:** 6
**Confidence:** 3

**Review:**

This paper seems to follow the guidelines for reproducibility, notation, code. The authors propose a method for counterfactual explanation on GNN which uses an extraction decision boundaries shared by multiple samples. They use empirical analysis to justify their theoretical proposal.

The author is working in the computational humor domain.  They collect demographic information from the annotators of humor datasets in an effort to learn more about the subjectivity inherent in the annotation process.  The author used the Prolific Academic platform to crowdsource annotations from users in young and old age groups.  The author noted a variation in labelling between age groups, older age groups finding items more offensive and less humorous, less likely to find something both offensive and humorous.

The author suggests the addition of an ’offensive’ label to reflect the fact a text as what may be humorous to one group, may be offensive to another. This would allow for more meaningful shared tasks and could lead to better performance on downstream applications, such as
content moderation.

The paper is largely about a shared assumption of datasets in what is otherwise a difficult field to apply NLP.  I do agree with this supposition but there is not a large amount of computational methods employed here, nevertheless there may be some researchers who find changes to existing dataset annotations, useful.

---

### Meta-Review · Area_Chair_qzCz · 2022-04-09

**Recommendation:** Reject
**Confidence:** 4

**Metareview:**

Reviewers remarked that the report is unclear in places, especially in that there seems to have been some human experiments, even though there were no detailed descriptions of how those were conducted.

---

### Decision · Program_Chairs · 2022-04-09

Reject